# Assault-related severe ocular chemical injury at a London ophthalmic referral hospital: a 3-year retrospective observational study

Jeremy John Hoffman  ,[1,2] Edward Joshua Casswell,[1,2] Alex John Shortt[2,3]

JJH and EJC contributed equally.

[1]Accident and Emergency Department, Moorfields Eye Hospital NHS Foundation Trust, London, UK
[2]National Institute of Health Research (NIHR) Biomedical Research Centre, University College London Institute of Ophthalmology, London, UK
[3]Department of External Eye Diseases, Moorfields Eye Hospital NHS Foundation Trust, London, UK

**Correspondence to**
Dr Jeremy John Hoffman;
jeremy.hoffman@lshtm.ac.uk

## ABSTRACT

**Objectives** To understand the incidence, causes, management and outcomes of intentional (assault) and unintentional severe ocular chemical injuries (SOCI) at an urban tertiary referral centre in the UK.

**Design** Retrospective observational study.

**Setting** A London tertiary referral ophthalmic centre, Moorfields Eye Hospital.

**Participants** All cases of SOCI presenting between 1 September 2011 and 31 August 2014 were identified. The definition of SOCI was grade 3 or 4 on the Hughes-Roper-Hall classification system. We identified 25 cases (6 in 2011–2012, 8 in 2012–2013, 11 in 2013–2014). Median age was 31.1 years. 23 cases (92%) were male.

**Primary and secondary outcome measures** The primary outcome was the proportion of cases of SOCI caused by assault, per year. Secondary outcome measures included the number of cases of SOCI, injury characteristics and mechanism, initial and long-term management, visual outcome and the need for surgical intervention.

**Results** Between 2011 and 2012, 3/6 cases were due to assault (50%); between 2012 and 2013, 7/8 were due to assault (87.5%); and between 2013 and 2014, 6/11 were due to assault (54.4%). Assault was responsible for 16/25 (64%) cases overall, while 8/25 (32%) cases were work related. The causative agent was known to be alkali in 16/25 (64%), while 10/25 (40%) did not complete the follow-up. The mean number of clock hours of limbal ischaemia was 5.24 (SD 2.97). 17/25 (68%) were Hughes-Roper-Hall grade 3. Surgical intervention occurred in 1/25. The final best-corrected visual acuity was 6/12 or worse in 11/25 (44%) and was counting fingers or worse in 4/25 (16%).

**Conclusions** Previous studies found that SOCI had a low incidence and that work-related injuries were the most common cause. Our study demonstrates an increasing incidence of SOCI, which may be accounted for by a rise in assault using corrosive substances. A high number of patients did not attend regularly for follow-up and visual outcomes from these injuries are poor.

## INTRODUCTION

Ocular chemical injury is an ophthalmic emergency and accounts for 11.5%–22.1% of ophthalmic trauma.[1] Rapid emergency first-aid management with copious irrigation is crucial, followed by appropriate acute ophthalmic

### Strengths and limitations of this study

► This is a case series, spanning 3 years, from a tertiary ophthalmic hospital in London.
► The primary outcome was the proportion of cases of severe ocular chemical injuries caused by assault, per year.
► The methodology differs to the previous national multicentre prospective (British Ophthalmological Surveillance Unit) study conducted between 2005 and 2006 meaning comparisons should be made with caution.

care.[2] Fortunately, most injuries are minor and unintentional.[3] However, severe ocular chemical injuries (SOCI), although infrequent, carry a high risk of developing devastating, long-term sequelae and sight loss if not managed appropriately and promptly.[2]

There are several factors that influence the severity of the injury, including the agent itself (acid or alkali), its strength, length of contact with the eye, volume and concentration. SOCI usually results from strong acid or alkali agents.[4] Alkali injuries are often most severe, as the basic charged molecules can rapidly cross the corneal epithelium and stroma and enter the anterior chamber.[2]

Several grading systems for ocular chemical injuries exist. Hughes noted the relationship between initial presenting clinical signs and prognosis; this was subsequently amended by Ballen and then Roper-Hall.[5–7] Although more recent systems have been developed, the Hughes-Roper-Hall (H-R-H) classification system is easy to use and remember, maintaining its popularity particularly in the ophthalmic emergency department.[8] The H-R-H system divides the injury into four grades. Grade 1 cases are mild with corneal epithelial damage only and no limbal ischaemia with a clear cornea. Grade 2 demonstrates a hazy cornea but with iris details still

visible and less than one-third (three clock hours) limbal ischaemia. Grade 3 involves between one-third and one-half limbal ischaemia (three to six clock hours), hazy cornea obscuring iris details and complete epithelial loss. Grade 4, the most severe, presents with an opaque cornea and greater than one-half limbal ischaemia. H-R-H grades 3 and 4 are considered SOCI.[3] The most recent evolution of burns classification is the Dua classification which includes an assessment of the degree of conjunctival burn involvement.[9]

A surveillance study of SOCI conducted in the UK from December 2005 to November 2006 only identified 12 cases this year, with a minimum estimated incidence of 0.02 per 100 000.[3] They found that 8/12 (66.7%) patients with SOCI were working-age men and half (6/12) occurred at work. Assault accounted for 4/12 (33.3%) of injuries.[3] These findings are supported by previous studies investigating mild, as well as SOCI.[1 4 10] In one study from the USA, similar demographics were reported, although assault only accounted for 2 out of 640 cases (0.31%).[11] A more recent study from the USA investigating the epidemiological trends of ocular chemical injuries (including mild cases) reports a median age of 32, with 56.6% of cases being male. Age-specific incidence rates per 100 000 population were highest among individuals aged between 20 and 29, with a mean of 17.4/100 000. This study also reported a high incidence among infants and children. The authors did not report whether injuries were intentional or unintentional.[12]

SOCI can either be caused by purposeful human action, defined as an 'intentional injury'; chemical assault is an example of this. Other injuries are those that are accidental, referred to as unintentional injuries. In the UK, particularly in metropolitan centres such as London, there has been a rapid rise in chemical assault or so-called 'acid' attacks.[13] This increase has received significant attention in the press with a number of high-profile attacks, with the incidence increasing by 65% in 2017 alone.[14] In London, the Metropolitan Police recorded an increase from 162 cases in 2012 to 454 in 2016; this number is higher than all the chemical assaults recorded in India that year.[13] However, a retrospective study from a leading national burns centre in London questions whether this increase that is being reported in the national press is translating into a similar increase in chemical burn presentations.[15]

The aim of this study was to investigate the number and causes of SOCI presentations per year over a 3-year period at Moorfields Eye Hospital, a central London tertiary referral eye hospital with a 24-hour ophthalmic emergency department. In addition, this study aimed to describe the demographics, responsible agent, clinical presentation and management of SOCI cases.

## MATERIALS AND METHODS

This study was a retrospective, observational study of patients attending Moorfields Eye Hospital Emergency Department with SOCI. Formal patient consent was deemed not necessary by the Moorfields ROAD Ethics Committee.

SOCIs were defined as either grade 3 or 4 on the H-R-H classification system.[5 7] The H-R-H classification was chosen over the newer Dua classification to follow the same inclusion criteria as reported in the British Ophthalmological Surveillance Unit (BOSU) study.[3] Furthermore, this classification is what is currently used and documented in our emergency department, with the treatment protocol dependent on the H-R-H grade.

All electronic records of emergency department attendances between 1 September 2011 and 31 August 2014 were searched for the following coded diagnoses 'Chemical Injury or Trauma' or 'Chemical Trauma' who were coded as either being admitted or followed up in clinic. Patient records were excluded from review if they were discharged directly from the department without follow-up as these cases would have represented minor injuries. All detailed clinical records of these attendances were then reviewed by one of two ophthalmologists (EJC and JJH) at Moorfields Eye Hospital. Cases meeting the SOCI definition above were included in the study. Patients with multiple attendances for the same initial event were only included once (ie, duplicate records were excluded). Patients without H-R-H 3 or 4 were excluded from the study; all other records were included. For any missing data or outlying values, the records were reviewed again by the other ophthalmologist to ensure quality control. Data entry and management was conducted within Microsoft Excel 2016 (Microsoft, Ridgemont, Virginia, USA), while statistical analysis was performed using STATA V.15 (StataCorp). To calculate p values to determine significance when comparing intentional to non-intentional injuries, means were compared using the t-test, while categorical variables were compared using $\chi^2$ calculations.

The primary outcome was the proportion of cases of SOCI caused by assault, per year. Secondary outcomes included the number of cases of SOCI, characteristics and mechanism of the injury, initial and long-term management, visual outcome and the need for surgical intervention.

### Patient and public involvement

Following the publication of our research in an open-access, peer reviewed journal, we plan on raising the awareness of the risks of chemical injury to the eye and measures to prevent this from happening, through public awareness campaigns. We would also seek to involve first-responders in disseminating the advice on the emergency treatment of ocular chemical injuries.

## RESULTS

There were 242 attendances coded as either 'Chemical Injury or Trauma' or 'Chemical Trauma' over the study period. Complete clinical records were available and reviewed for 219 individuals, while the electronic records were available for 19 cases. There were three duplicate

| Table 1 Patient demographics, mechanism, place of injury and chemical agent responsible for the injury | | |
|---|---|---|
| | **n** | **(%)** |
| **Patients** | 25 | 100 |
| Male | 23 | 92 |
| Female | 2 | 8 |
| **Mechanism and place of injury** | | |
| Intentional-assault | 16 | 100 |
| Public-street | 8 | 50 |
| Public-public transport | 1 | 6.25 |
| Public-bar/pub | 1 | 6.25 |
| Work | 1 | 6.25 |
| Home | 2 | 12.5 |
| Unknown | 3 | 18.75 |
| Unintentional | 9 | 100 |
| Work-cleaning | 3 | 33.3 |
| Work-building | 3 | 33.3 |
| Work-vehicle | 1 | 11.1 |
| Home-cleaning | 1 | 11.1 |
| Home-other | 1 | 11.1 |
| **Chemical agent** | 25 | 100 |
| Acid | 1 | 4 |
| Alkali | 16 | 64 |
| Other | 2 | 8 |
| Unknown | 6 | 24 |

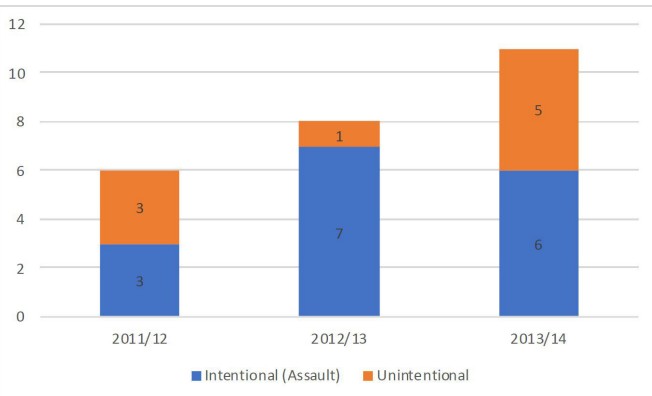

**Figure 1** Patients with assault-related severe ocular chemical injury presenting to Moorfields eye Hospital between 2011 and 2014 as a proportion of severe ocular chemical injury presentations.

81 899 in 2012–2013 and 85 799 in 2013–2014. This gives a presentation rate of SOCI at Moorfields of 0.773 cases per 10 000 encounters in 2011–2012; 0.977 cases per 10 000 encounters in 2012–2013; and 1.28 cases per 10 000 encounters in 2013–2014. There was a mean year-on-year increase of 0.254 cases per 10 000 encounters. There was an approximate increase of 65.6% SOCI encounters from 2011–2012 to 2013–2014.

### Clinical presentation

Tables 2 and 3 describe the baseline and follow-up visual acuity, H-R-H grade, surgical management and clinical outcome for our cohort of patients. At presentation, the visual acuity was 6/18 (20/60) or worse in 15/25 (60%) of patients, with 3/25 (12%) of patients 6/60 (20/200) or worse. The mean number of clock hours of limbal ischaemia was 5.24 (SD 2.97). There were 17/25 (68%) cases of H-R-H grade 3 and 8/25 (32%) cases of H-R-H grade 4. Time to irrigation data were available in 9/25 (36%) cases: irrigation was performed within 1 hour of the index event in only 5/9 (55%) of these cases, while the mean time to irrigation was 2.6 hours (range 0.5–15 hours; SD 4.70).

### Management

The initial management instigated in our emergency department is described in figure 2. All patients received a topical antibiotic and steroid at presentation, 22/25 (88%) of whom received preservative-free formulations. Topical citrate and ascorbate were prescribed in 18/25 (72%) and 21/25 (84%) of cases, respectively. Oral ascorbate and oral doxycycline were administered to 19/25 (76%) and 15/25 (60%) of patients, respectively. The full recommended protocol of preservative free antibiotics and steroids; topical citrate and ascorbate; and oral ascorbate and doxycycline was followed in 5/25 (20%).

### Long term management and outcome

Of the 25 patients with H-R-H grade 3–4 injuries, 10/25 (40%) failed to comply with standard follow-up regimens in our department and were lost to follow-up at various

records and one record was unavailable or missing. The retrieval rate for the study was, therefore, 99.6%. Of the 238 notes reviewed, 25 cases met the inclusion criteria of SOCI.

### Demographics and mode of injury

The median age at presentation was 31.1 years, (range 17–74, IQR 27–42). Table 1 describes the patient demographics and mode of injury, where 23/25 cases (92%) were male. Intentional injury caused by assault was responsible for 16/25 (64%) cases across the 3 years, with the remaining 9/25 (36%) classified as unintentional or accidental. Of these, 7/9 (77.8%) of the cases were work related. Intentional injuries occurred predominantly in public places (including the street, public transport and bars/pubs), accounting for 10/16 (62.5%) injuries. Intentional injuries at home and at work occurred in 2/16 (12.5%) and 1/16 (6.25%) of cases, respectively. Injury with an alkali substance was reported in 16/25 (64%) and unknown in 6/25 (24%).

There were 6 cases of SOCI identified between 2011 and 2012, of which 3/6 were due to assault (50%); 8 between 2012 and 2013, 7/8 due to assault (87.5%) and 11 between 2013 and 2114, 6/11 due to assault (54.4%) (figure 1). During the same period, there were 77 652 encounters in the emergency department in 2011–2012;

**Table 2** Best-corrected visual acuity (BCVA) at presentation and at final follow-up, Hughes-Roper-Hall (H-R-H) grade, surgical intervention and corneal sequelae for all patients, and whether patients were lost to follow-up

| Patient | BCVA at presentation | BCVA at final follow-up | H-R-H grade | Surgical intervention | Long-term complications | Lost to follow-up? |
|---|---|---|---|---|---|---|
| 1 | 6/9 | 6/4 | 3 | – | | No |
| 2 | 6/18 | 6/18 | 3 | – | KS, CCS | No |
| 3 | 6/24 | 6/6 | 3 | – | | No |
| 4 | 6/6 | 6/9 | 3 | – | CCS | No |
| 5 | 6/12 | 6/5 | 3 | – | | DNA at 1/52 |
| 6 | 6/6 | 6/9 | 3 | – | KS | No |
| 7 | 6/24 | Unknown | 3 | – | | DNA at 1/52 |
| 8 | 6/9 | 6/12 | 4 | – | | DNA at 1/12 |
| 9 | 6/9 | 6/5 | 3 | – | | No |
| 10 | 6/24 | 6/6 | 4 | – | KS, CCS, CP | No |
| 11 | 6/9 | 6/18 | 4 | – | | DNA at 1/12 |
| 12 | 6/18 | 6/6 | 3 | – | CCS | No |
| 13 | 6/12 | 6/12 | 3 | – | | DNA at 1/12 |
| 14 | 6/36 | 6/9 | 4 | – | | DNA at 1/12 |
| 15 | 6/24 | 6/12 | 4 | – | | DNA at 1/12 |
| 16 | 6/18 | 6/9 | 3 | – | KS, CCS | No |
| 17 | CF | HM | 4 | – | PED | No |
| 18 | 6/18 | 6/12 | 3 | – | | No |
| 19 | 4/60 | 3/60 | 3 | – | | DNA at 3/12 |
| 20 | 6/12 | HM | 3 | – | KS, PED, CCS | DNA at 12/12 |
| 21 | 6/36 | HM | 4 | – | KS, PED, MK, CCS, CP, Ent | No |
| 22 | 6/36 | 6/6 | 3 | – | | DNA at 1/12 |
| 23 | 6/12 | 6/5 | 3 | – | | No |
| 24 | 3/60 | 6/12 | 4 | – | | No |
| 25 | 6/18 | 6/9 | 3 | AMG, LSC, PK, Phaco | PED, CCS, Cat | No |

AMG, amniotic membrane graft; CCS, central corneal scar; CF, counting fingers; CP, central pannus; DNA, did not attend follow-up; Ent, entropion; HM, hand movements; KS, keratoconjunctivitis sicca; LSC, limbal stem cell graft; MK, microbial keratitis; PED, persistent epithelial defect; Phaco, phacoemulsification and intra-ocular lens insertion; PK, penetrating keratoplasty.

stages of treatment. Of 10, two patients did not attend (DNA) for review beyond 1 week and were discharged, 6/10 DNA reviews beyond 1 month and 2/10 failed to attend appointments between 3 and 12 months, despite being offered alternative appointments, as per hospital policy. The final best-corrected visual acuity (defined here as the later of either the last clinic appointment in the patient records prior to discharge, or 1 year following the index event) was less than 6/12 (20/40) in 11/25 (44%) of cases. There were 4/25 (16%) of patient with vision at counting fingers or worse at final follow-up (tables 2 and 3).

Long-term complications included keratoconjunctivitis sicca (6/25), persistent epithelial defects (4/25), microbial keratitis (1/25), central pannus (2/25), central corneal scarring (8/25), entropion (1/25) and cataract (1/25) (table 2). Of note, there were no recorded cases of corneal melt, perforation, secondary glaucoma or patients requiring eye removal.

Surgical intervention was performed in 1/25 (4%), who underwent an amniotic membrane graft, a limbal stem cell graft and finally a penetrating keratoplasty. This same patient subsequently developed cataract and underwent uncomplicated phacoemulsification and intraocular lens insertion.

**Comparison between intentional and unintentional injuries**

Table 3 compares the demographics, management and outcomes for patients whose injuries were classified as either intentional (ie, assault) or unintentional. Intentional injuries occurred solely among men, although this was not found to be statistically significant using Fisher's exact test (p=0.12), likely due to the small sample size. Unintentional injuries occurred solely at home or at

**Table 3** Comparison of select patient demographics, injury mechanism, place of injury, clinical features, management and outcome between patients sustaining an intentional injury (ie, assault) and those sustaining an unintentional (accidental) injury

| | Intentional (assault) n=16 | | Unintentional n=9 | | P value |
|---|---|---|---|---|---|
| | n | (%) | n | (%) | |
| **Patients** | 16 | 100 | 9 | 100 | |
| Male | 16 | 100 | 7 | 77.8 | 0.12 |
| Female | 0 | 0 | 2 | 22.2 | |
| **Median age (years)** | 29.7 | | 38.3 | | 0.0897 |
| **Mechanism and place of injury** | | | | | |
| Injury occurred at home or work | 3 | 18.75 | 9 | 100 | <0.001* |
| Injury occurred in public | 10 | 62.5 | 0 | 0 | 0.003* |
| Alkali injury | 10 | 62.5 | 6 | 66.7 | 1 |
| Acid injury | 0 | 0 | 1 | 6.25 | 0.36 |
| **Presenting clinical features** | | | | | |
| Presenting BCVA ≤6/12 | 10 | 62.5 | 5 | 55.6 | 1 |
| Presenting BCVA ≤/60 | 3 | 18.8 | 0 | 0 | 0.28 |
| Mean limbal ischaemia | 5.75 (SD 2.26) | | 4.33 (SD 3.94) | | 0.8293† |
| H-R-H grade 3 | 10 | 62.5 | 7 | 77.8 | 0.661 |
| H-R-H grade 4 | 6 | 37.5 | 2 | 22.2 | 0.661 |
| **Management** | | | | | |
| Mean time to irrigation (hours) | 5.67 (SD 8.09) | | 1.17 (SD 0.98) | | 0.7819† |
| Preservative free antibiotics and steroids | 12 | 75 | 7 | 77.8 | 1 |
| Full treatment protocol | 2 | 12.5 | 3 | 33.3 | 0.312 |
| **Outcome** | | | | | |
| Patient failed to attend follow-up | 6 | 37.5 | 4 | 44.4 | 1 |
| Surgical intervention | 1 | 6.25 | 0 | 0 | 1 |
| Vision ≤6/12 | 7 | 43.75 | 4 | 44.4 | 1 |
| Vision ≥CF | 3 | 18.75 | 1 | 11.1 | 1 |

*Statistically significant p value. Statistical testing performed: t-test for means if normally distributed; if variances are unequal then adjusted using Welch's formula.
†Fisher's exact for categorical variables.
BCVA, best corrected visual acuity; H-R-H, Hughes-Roper-Hall Grade.

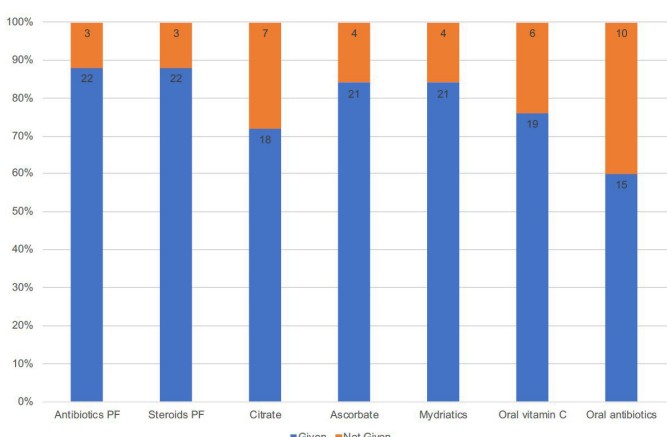

**Figure 2** Initial acute management given in the emergency department for patients presenting with severe ocular chemical injury (n=25) during the 3-year study period. PF, preservative free.

work, while the majority of intentional injuries occurred in public places (10/16, 62.5%). These differences in location between the intentional and unintentional injury groups was found to be statistically significant (p=0.003 and p<0.001 for injury in public and at home or work, respectively). Other than these, there was no statistically significant difference identified between the intentional and unintentional injury groups in terms of presenting features, management or outcome.

## DISCUSSION

The number of cases of SOCI presenting to Moorfields Eye Hospital increased year-on-year between 2011 and 2014, at a mean increase of 0.254 cases per 10 000 encounters per year. There was an increase of 65.6% from 2011/2012 to 2013/2014. The number of cases attributable to assault doubled from 3/6 (50%) to 7/8 (87.5%) between 2011/2012 and 2012/2013. In 2013/2014, there

were 6/11 cases of SOCI caused by assault. Overall across the 3 years, assault was responsible for 16/25 (64%) presentations.

These results contrast to those of the BOSU study from 2005 to 2006.[3] The calculated incidence in 2005 was 0.02 per 100 000, based on the 12 patients reported over the course of the year and assuming a UK population of 60 million. We report that in 2013/2014, there were 11 cases of SOCI seen within one tertiary eye unit in Central London. Assuming that Moorfields Eye Hospital serves half of the London population for ophthalmic services and a population of 8.79 million, we estimate that the incidence of SOCI is 0.25 per 100 000 in London. This is more than 10 times greater than the 2005–2006 study. Furthermore, we have found that the proportion of injuries due to assault compared with work related is 64%–28%, respectively; this contrasts to the BOSU results of 33% and 50%, respectively. Demographic characteristics between the two studies are similar with SOCI being more common in young males, although we found a greater proportion of males (92% compared with 75%).

The authors of the BOSU study anticipated a maximum of 200–300 cases of SOCI annually in the UK, based on two previous studies.[3 4 10] If our results were extrapolated across the UK with a population of 65.6 million, we would expect approximately 164 case of SOCI annually. This is closer to Macdonald *et al's* anticipated level of cases. However, it must be emphasised that the methodology differs significantly between the two studies: one is a retrospective review of cases at a single tertiary centre while the other is a prospective survey conducted across the UK. This means that direct comparisons and any conclusions should be made with caution. It is also likely that the BOSU study may have under-reported the results by up to 25%.[3] With our study being conducted in a tertiary centre, our results may overestimate the incidence. Furthermore, it is likely that chemical injuries due to assault are higher in cities, and possibly in London, compared with the rest of the UK. We chose to conduct this present study given anecdotal evidence of an increase in SOCI at our centre. As opposed to the prospective methodology used in BOSU surveillance studies, this enables a 'snapshot' to gauge trends. We would, therefore, advocate further nationwide multicentre prospective studies to evaluate the true incidence of SOCI.

A recent study by Bizrah and coworkers from our same institution reported 10 patients with SOCI identified over 3 months in 2016, of which seven (70%) were attributable to assault.[16] This suggests that the incidence of SOCI continues to increase and that assault-related chemical injuries continue to be a problem, with a similar proportion of assault-related injuries as in our cohort of patients.

The reasons behind the rise in 'acid attacks' are likely multifactorial. In the past, acid attacks were previously related to robberies. Now, with tougher legislation against carrying knives, corrosive substances were seen as a replacement weapon that was harder to detect, with no legislation previously in the UK against carrying it.

However, the Offensive Weapons Act 2019, closes this loophole and places restricitons on the sale of corrosive substances and makes it an offensive to have a corrosive substance in a public place.[17] Furthermore, a person found guilty is liable for a fine and/or imprisonment for up to 12 months. This is in addition to sentencing guidelines that classify corrosive substances as 'highly dangerous weapons', meaning longer custodial sentences for those found guilty of using a corrosive substance to threaten or assault others.[18] Many of these attacks are related to gangs, with men being twice as likely as women to be the victims.[13] It will be interesting to see if these legislative changes will translate to a reduced incidence of assault-related SOCI and further studies similar to ours and that of Bizrah and coworkers are advocated.[16]

An interesting result from our study is that 40% (10/25) of our patients were lost to follow-up as they DNA subsequent review appointments. This raises several observations. First, this may affect the results of our study. For example, the final visual acuity may be better than we reported if patients with better vision were more likely to DNA. Second, although this DNA rate is the same as that reported from a recent study relating to SOCI,[16] this rate is considerably higher than that reported by previous studies of general ophthalmic outpatient attendances in the UK where the reported rate is 12.5%–12.9%.[19 20] This is likely to be due to demographics; DNA rates are typically higher in young men, those from deprived backgrounds and those living in urban areas.[21] This patient demographic is likely over-represented in our study population, compared with the general ophthalmic population. When comparing intentional to unintentional injuries, there was no statistically significant difference between the DNA rates between the groups (table 3), indicating that assault itself was not a predictor for patients who eventually DNA. This finding has important implications for outpatient management: patients attending with SOCI should receive additional counselling on the importance of their follow-up appointments and may benefit from additional reminders via telephone calls, emails or text messages.

The most important step to treating chemical injury and improving the outcome is immediate irrigation—'dilution is the solution'.[1 4 22] This is ideally done immediately at the scene following the exposure, continued by the paramedical team in the ambulance on the way to hospital and repeated again at presentation until the pH has normalised. This should be performed with normal saline where available, but if this is not available, clean water should be used. In our study, this was performed in 5/9 (55%) of patients within 1 hour of the injury, less than the 75% reported by Macdonald *et al*, although it is important to consider that we only had data for 36% participants.[3] There is therefore a need to improve people's awareness of the importance of irrigation. Given the shift in the mode of injury from workplace-related injury, for which there already exists robust health and safety legislation and awareness, to assault related, we

suggest raising awareness among people at risk of falling victim to 'acid attacks' through appropriately targeted advertising campaigns.

The aim of early medical management is to reduce inflammation and promote corneal re-epithelialisation, and prevent acute complications such as infection or corneal melt.[2] Treatment should avoid the use of preserved medications, particularly those containing phosphates, as this can precipitate acute corneal calcification.[23 24] Acute treatment therefore involves preservative-free topical steroids and antibiotics to reduce inflammation and prevent infection. Topical citrate and ascorbate helps to prevent corneal melts.[25] Oral doxycycline can reduce the inflammation and the risk of melt.[2] Oral ascorbate 1 g daily can also prevent collagen lysis, promote healing and prevent corneal melts.[2] The protocol at Moorfields Eye Hospital is that all patients with SOCI receive preservative-free steroid and antibiotic, topical ascorbate and citrate, oral ascorbate and doxycycline and cycloplegia. In our study, while all patients received topical steroid and antibiotics, 76% (19/25) received preservative-free formulation. This has improved since the BOSU study where only half (6/12) of patients received preservative-free formulations.[3] Topical citrate and oral doxycycline were prescribed in 48% of patients (12/25), compared with 25% (3/12) of patients in the BOSU study.[3]

Surgical intervention was low in our study, with only one patient (1/25, 4%) undergoing surgical treatment. This same patient underwent multiple forms of ocular surface reconstructive surgery. This low surgical rate of intervention seen in our study compares to 50% (6/12) of cases in the BOSU study, of which half of these had more than one surgical procedure.[3] Reasons for this are unclear, but it is possible that the high DNA rate may have resulted in a falsely low rate of surgical intervention, with patients being lost to follow-up before surgical intervention was appropriate.

Final vision was reduced to worse than 6/12 in 44% of cases, which is marginally better compared with 50% reported by Macdonald *et al*.[3] The rate of very severe visual loss (worse than 6/60) was 16% in our study, comparable to 16.7% in the BOSU study.[3] This suggests that on the whole, visual prognosis from SOCI remains poor, with the potential for significant morbidity and ocular sequelae as a result of the injury.

There are a number of limitations to our study. First, as mentioned above, the methodology between our study and that of the BOSU study are different, meaning direct comparisons should be interpreted with a degree of caution. Second, although we report a large case series relative to other other studies, the absolute number of patients in our study, despite it running for 3 years, is only 25, as SOCI is a rare condition. This may make statistical analysis challenging, as our sample size may be insufficiently powered to detect significance. Third, with this study being a retrospective review of case notes,

the accuracy of the data collected was dependent on the record-keeping of the attending clinician. This was minimised by cross-checking the paper clinical records with online records that include patient letters and surgical interventions, to ensure we did not miss any outcome data. We are, therefore, confident that we were able to gather all the relevant clinical data for this cohort. Finally, the data from this study are from 2011 to 2014 and may not necessarily be representative of what is currently being seen, particularly given there have been new legislative changes since this data was collected, introduced in 2019.[17] Until that date, intentional injuries caused by corrosive substances had been increasing, suggesting our results represent the start of this becoming a significant problem in the UK and in particular the London region. This is supported by the results form a London-based study that reported 70% of SOCI were caused by intentional injury in 2016.[16] Data on the number of assaults caused by corrosive substances since the legislative changes have yet to be published; once this is available a subsequent study looking at SOCI at our institution would be helpful to see the current trend.

This study suggests that there has been a recent increase in the incidence of SOCI in a central London tertiary eye hospital, with assault now responsible for between one half and just under two-thirds of cases. It is likely that SOCI is more common than has previously been reported, particularly in metropolitan areas. This increase may be a result of the recent increase in 'acid attacks', although further work is necessary to establish causality. Regardless, there is a need to reverse the increase in 'acid' attacks through stronger legislation to act as a deterrent. There was a lower than expected level of initial irrigation and so public awareness of how to manage chemical injury could be improved. Similarly, first responders and emergency department personnel should be reminded of the importance of copious irrigation and onward referral to a specialist ophthalmic unit. Ophthalmologists attending these cases should follow an evidenced-based protocol and be aware of the risk of patients not attending their follow-up appointments and plan accordingly. An updated nationwide surveillance study is warranted to enable direct comparisons to what was seen nearly 15 years ago.

**Contributors** Searched the literature: JJH, EJC and AJS. Drafted initial protocol: EJC. Gained ethical approval: EJC. Contributed to protocol development and revision: JJH, EJC and AJS. Database design: EJC. Data collection: JJH and EJC. Statistical analysis: JJH. Database management: JJH. Drafted this manuscript: JJH. Critically revised this manuscript: JJH, EJC and AJS. Conceptualisation: EJC and AJS. Supervision: AJS, JJH and EJC contributed equally to the preparation of this manuscript.

**Funding** JJH was supported by the National Institute for Health Research (NIHR) Biomedical Research Centre based at Moorfields Eye Hospital NHS Foundation Trust and UCL Institute of Ophthalmology (ACF-2015-18-020). AJS was supported by a Wellcome Trust Intermediate Clinical Fellowship based at the UCL Institute of Ophthalmology and the UCL Institute of Immunity and Transplantation (102804/Z/13/Z). EJC was supported through a grant from the Friends of Moorfields Eye Hospital Charity.

**Competing interests** None declared.

**Patient consent for publication** Not required.

**Ethics approval** The study received approval from the Moorfields Eye Hospital Research on Anonymised Data (ROAD) Ethics Committee in February 2015 (Ref: ROAD 15/010) and the Moorfields Eye Hospital Clinical Audit Department (Ref: CA15/CED/11). The study was compliant with the Declaration of Helsinki.

**Provenance and peer review** Not commissioned; externally peer reviewed.

**Data availability statement** Data are available on reasonable request. There are no additional unpublished data from this study. The datasets generated during and/or analysed during the current study will be available upon request from JJH (jeremy.hoffman@lshtm.ac.uk). The full data set will be available with all patient identifiable details removed. Datasets will only be available to bona fide scientific investigators. Requests should be made to JJH in writing detailing the scientific investigators background and intended use for the data. Consideration will be given to all proposed analyses.

**ORCID iD**
Jeremy John Hoffman http://orcid.org/0000-0001-9454-2131

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
