## [Reviewer comments · BMJ Open]

ARTICLE DETAILS

TITLE (PROVISIONAL)	Assault-related severe ocular chemical injury at a London ophthalmic referral hospital: a three-year retrospective observational study
AUTHORS	Hoffman, Jeremy John; Casswell, Edward J; Shortt, Alex

VERSION 1 – REVIEW

REVIEWER	Francisco Javier Bonilla Escobar University of Pittsburgh, USA Univesidad del Valle, Colombia
REVIEW RETURNED	14-Apr-2020

GENERAL COMMENTS	Thank you for allowing me to review this manuscript. Ocular burns are an important public health issue and there is a lack of attention to it internationally and big gaps of knowledge about basic statistics of the problematic worldwide. The paper is well-written and the methods are reliable. There are some aspects that I will like to suggest to review: 1. Abstract: some conclusions are described but not presented in the results.2. Introduction: lacks of a description of the proper terminology of injuries: intentional and no intentional (unintentional)3. Methods: there is missing information about the data collection, how, who and where the data was collected? There are not exclusion criteria mentioned. There is no mention of the quality control of the data or how were handled the missing values or extreme values. The statistical analysis lacks the univariate and bivariate description. The multivariable model is unreliable due to the sample size so I will suggest to remove it.5. In the result section, I will suggest the authors to describe also the unintentional injuries. Additionally, a comparison between intentional and unintentional injuries in terms of demographics, management and outcomes will enhance the results of the study.6. Discussion: The limitations are not clearly stated and the article is exaggerating the results when compared with a national study or other studies in variables with too many missing values such as irrigation.7. The references are more than 10 years old which is understandable in a topic that doesn't have much published; however, the authors need to make the extra effort to update their references.8. Figures: pie diagrams are not common in medical literature. Please change them to bar diagrams. Other comments can be found in the attached pdf. Thank you again.
---

VERSION 1 – AUTHOR RESPONSE

Reviewer 1

Thank you for allowing me to review this manuscript. Ocular burns are an important public health issue and there is a lack of attention to it internationally and big gaps of knowledge about basic statistics of the problematic worldwide. The paper is well-written, and the methods are reliable. There are some aspects that I will like to suggest to review:

1. Abstract: some conclusions are described but not presented in the results.

Thank you for this comment. We have removed the reference to irrigation from the conclusion so now all the conclusions mentioned should relate to the results given in the abstract.

2. Introduction: lacks of a description of the proper terminology of injuries: intentional and no intentional (unintentional)

Thank you for this comment. We have added the following sentence to the beginning of the penultimate paragraph of the introduction:

SOCI can either be caused by purposeful human action, defined as an “intentional injury”; chemical assault is an example of this. Other injuries are those that are accidental, referred to as unintentional injuries.

3. Methods: there is missing information about the data collection, how, who and where the data was collected? There are not exclusion criteria mentioned. There is no mention of the quality control of the data or how were handled the missing values or extreme values. The statistical analysis lacks the univariate and bivariate description. The multivariable model is unreliable due to the sample size so I will suggest to remove it.

Thank you for this comment. I have re-written the third paragraph of the methods section to address these questions. We have removed the section describing the statistical analysis as advised and updated it with the statistical analysis performed for the comparison between intentional and unintentional injuries. This section now reads:

All electronic records of emergency department attendances between 1st September 2011 and 31st August 2014 were searched for the following coded diagnoses “Chemical Injury or Trauma” or “Chemical Trauma” who were coded as either being admitted or followed-up in clinic. Patient records were excluded from review if they were discharged directly from the department without follow-up as these cases would have represented minor injuries. All detailed clinical records of these attendances were then reviewed by one of two ophthalmologists (EC and JH) at Moorfields Eye Hospital. Cases meeting the SOCI definition above were included in the study. Patients with multiple attendances for the same initial event were only included once (i.e. duplicate records were excluded). Patients without H-R-H 3 or 4 were excluded from the study; all other records were included. For any missing data or outlying values, the records were reviewed again by the other ophthalmologist to ensure quality control. Data entry and management was conducted within Microsoft Excel 2016 (Microsoft Corp., Ridgmont, VA), whilst statistical analysis was performed using STATA 15 (StataCorp, College Station, TX). To calculate p-values to determine significance when comparing intentional to non-intentional injuries, means were compared using the t-test, whilst categorical variables were compared using chi-squared calculations.

5. In the result section, I will suggest the authors to describe also the unintentional injuries.

Additionally, a comparison between intentional and unintentional injuries in terms of demographics, management and outcomes will enhance the results of the study.

Thank you for this suggestion. We have now added a new section, Comparison between intentional and unintentional injuries, together with a new Table 3, that compares the results between intentional and unintentional groups. We feel that this improves the overall quality of the paper. This section reads as follows:

Comparison between intentional and unintentional injuries

Table 3 compares the demographics, management and outcomes for patients whose injuries were classified as either intentional (i.e. assault) or unintentional. Intentional injuries occurred solely amongst men: this was found to be statistically significant when compared to unintentional injuries ($p = 0.049$). Unintentional injuries occurred solely at home or at work, whilst the majority of intentional injuries occurred in public places (10/16, 62.5%). These differences in location between the intentional and unintentional injury groups was found to be statistically significant ($p < 0.001$). Other than these, there was no statistically significant difference identified between the intentional and unintentional injury groups in terms of presenting features, management or outcome.

6. Discussion: The limitations are not clearly stated, and the article is exaggerating the results when compared with a national study or other studies in variables with too many missing values such as irrigation.

Thank you for this comment. We have now added a paragraph that details the limitations of this study. This mentions that one must be cautious when comparing the results from our study with the BOSU National Surveillance Study. This new section reads as follows:

There are a number of limitations to our study. Firstly, as mentioned above, the methodology between our study and that of the BOSU study are different, meaning direct comparisons should be interpreted with a degree of caution. Secondly, although we report a large case series relative to other other studies, the absolute number of patients in our study, despite it running for three years, is only 25, as SOCI is a rare condition. This may make statistical analysis challenging, as our sample size may be insufficiently powered to detect significance. Thirdly, with this study being a retrospective review of case notes, the accuracy of the data collected was dependent on the record-keeping of the attending clinician. This was minimised by cross-checking the paper clinical records with online records that include patient letters and surgical interventions, to ensure we did not miss any outcome data. We are therefore confident that we were able to gather all the relevant clinical data for this cohort.

7. The references are more than 10 years old which is understandable in a topic that doesn't have much published; however, the authors need to make the extra effort to update their references.

Thank you for this comment. We have now repeated the literature search and added the references below to our manuscript and updated the text where relevant. We feel that this has significantly improved our manuscript.

Haring RS, Sheffield ID, Channa R, et al. Epidemiologic Trends of Chemical Ocular Burns in the United States. *JAMA Ophthalmol* 2016;134:1119–24. doi:10.1001/jamaophthalmol.2016.2645

Miller R, Pywell S, Leon-Villapalos J, et al. Chemical burn assaults: Is the media coverage representative? *Burns* 2018;44:1021–2. doi:10.1016/j.burns.2018.01.004

Bizrah M, Yusuf A, Ahmad S. Adherence to Treatment and Follow-Up in Patients with Severe Chemical Eye Burns. *Ophthalmology and Therapy* 2019;8:251–9. doi:10.1007/s40123-019-0173-y

8. Figures: pie diagrams are not common in medical literature. Please change them to bar diagrams.

Thank you. We have changed Figure 1 so that the data from this is now presented in Table 1. We have replaced Figure 1 with a new Figure 1 that describes the increasing incidence of SOCI at our institution over the study period, with the proportion of intentional and unintentional injuries shown.

We have also updated Figure 2 so that the formatting matches that of the new Figure 1.

Other comments can be found in the attached pdf. Thank you again.

Thank you for these comments, we have addressed these comments in our revision. We have also made a number of further amendments throughout the manuscript, updating some of the references as advised and correcting some errors that we have identified. We hope that this updated paper addresses your comments to your satisfaction.

VERSION 2 – REVIEW

REVIEWER	Francisco Javier Bonilla-Escobar University of Pittsburgh, USA. Universidad del Valle, Colombia.
REVIEW RETURNED	13-Aug-2020

GENERAL COMMENTS	Thank you for addressing the comments. The manuscript looks stronger. Two minor comments:  1. The age of the data needs to be discussed and how those injuries could have changed in recent years based on trends, legislation, etc. 2. It looks like Chi2 is not the proper test to compare the categories based on the number of observations. Before using this test you need to check the expected values. In Stata: tab var1 var2, exp Check if you have any expected value below 5 and for that analysis use the Fisher's exact test: tab var1 var2, exact Same for the t-test. Assumptions must be checked before running it. Normal distribution and equal variances (in Stata: swilk and sdtest). 2.1. The table just need one p-value per group of variables, i.e., sex vs intentionality Characteristic Intentionality P-value Male n (%) n (%) x.x Female n (%) n (%)
--

VERSION 2 – AUTHOR RESPONSE

Reviewer 1

Thank you for addressing the comments. The manuscript looks stronger. Two minor comments:

1. The age of the data needs to be discussed and how those injuries could have changed in recent years based on trends, legislation, etc.

Thank you for this comment. We have added a discussion around the age of the data and possible changes following legislation to the paragraph discussing the study's limitations within the Discussion section. This additional text reads as follows:

Finally, the data from this study is from 2011-2014 and may not necessarily be representative of what is currently being seen, particularly given there have been new legislative changes since this data was collected, introduced in 2019.[17] Until that date, intentional injuries caused by corrosive substances had been increasing, suggesting our results represent the start of this becoming a significant problem in the UK and in particular the London region. This is supported by the results from a London-based study that reported 70% of SOCI were caused by intentional injury in 2016.[16] Data on the number of assaults caused by corrosive substances since the legislative changes have yet to be published; once this is available a subsequent study looking at SOCI at our institution would be helpful to see the current trend.

2. It looks like Chi2 is not the proper test to compare the categories based on the number of observations. Before using this test you need to check the expected values.

In Stata:

tab var1 var2, exp

Check if you have any expected value below 5 and for that analysis use the Fisher's exact test:

tab var1 var2, exact

Same for the t-test. Assumptions must be checked before running it. Normal distribution and equal variances (in Stata: swilk and sdtest).

Thank you for this helpful comment relating to the statistical testing and for the Stata code. We have now updated Table 3 with values calculated using the Fisher's exact test for categorical variables, as all these variables had expected values of less than 5 making the Chi squared test inappropriate. For the continuous variables, we have used the t-test where the assumptions around normal distribution and equal variance are met. When these are not met, the calculations are adjusted using the Welch formula. The form of testing used has been indicated in the legend of Table 3. We also tried the Mann-Whitney Test (as a test for non-parametric data) for these variables and this also failed to give a statistically significant result. We hope that this is acceptable to the reviewer. An updated version of Table 3, together with the legend, is given below. We have also updated any sections within the body of the text that referred to Table 3 so that the quoted p-values mirror what is in the updated table.

2.1. The table just need one p-value per group of variables, i.e., sex vs intentionality

Characteristic Intentionality P-value

Male n (%) | n (%) x.x

Female n (%) | n (%)

Thank you for this comment, this has been corrected as indicated in the updated Table 3 below.

Table 3: Comparison of select patient demographics, injury mechanism, place of injury, clinical features, management and outcome between patients sustaining an intentional injury (i.e. assault) and those sustaining an unintentional (accidental) injury

	Intentional (assault) n = 16		Unintentional n = 9		p-value
	n	(%)	n	(%)	
Patients	16	(100)	9	(100)	
Male	16	(100)	7	(77.8)	0.12
Female	0	(0)	2	(22.2)	
Median age (years)	29.7		38.3		0.0897
Mechanism and place of injury					
Injury occurred at home or work	3	(18.75)	9	(100)	<0.001*
Injury occurred in public	10	(62.5)	0	(0)	0.003*
Alkali injury	10	(62.5)	6	(66.7)	1.00

Acid injury	0	(0)	1	(6.25)	0.360
Presenting clinical features					
Presenting BCVA <= 6/12	10	(62.5)	5	(55.6)	1.00
Presenting BCVA <= 6/60	3	(18.8)	0	(0)	0.280
Mean limbal ischaemia	5.75 (SD 2.26)		4.33 (SD 3.94)		0.8293 ‡
H-R-H Grade 3	10	(62.5)	7	(77.8)	0.661
H-R-H Grade 4	6	(37.5)	2	(22.2)	0.661
Management					
Mean time to irrigation (hours)	5.67 (SD 8.09)		1.17 (SD 0.98)		0.7819 ‡
Preservative free antibiotics and steroids			7	(77.8)	1.00
Full treatment protocol	12	(75)	3	(33.3)	0.312
Outcome					
Patient failed to attend follow-up			4	(44.4)	1.00
	6	(37.5)			
Surgical intervention	1	(6.25)	0	(0)	1.00
Vision <= 6/12	7	(43.75)	4	(44.4)	1.00
Vision <= CF	3	(18.75)	1	(11.1)	1.00

BCVA, Best Corrected Visual Acuity; H-R-H, Hughes-Roper-Hall Grade; SD, Standard Deviation; * =

statistically significant p-value. Statistical testing performed: t-test for means if

normally distributed; if variances are unequal then adjusted using Welch's formula

(‡); Fisher's exact for categorical variables.